# Simulation Research on Auxiliary Power Supply System of China Standard EMU

**Liwei Zhang \*** **, Lansi Yun, Minghe Sun and Bo Peng**

School of Electrical Engineering, Beijing Jiaotong University, Beijing 100044, China; 17121529@bjtu.edu.cn (L.Y.); 16117410@bjtu.edu.cn (M.S.); 18126132@bjtu.edu.cn (B.P.)

**\*** Correspondence: lwzhang@bjtu.edu.cn; Tel.: +86-10-51682335

**Abstract:** The auxiliary power supply system is an important part of the China standard EMU (Electric Multiple Units). It is mainly composed of auxiliary converters, chargers, battery packs and several loads. According to the topology of the EMU real auxiliary power supply system, the whole simulation system including the auxiliary converter, charger, single-phase inverter and other single models is built, and the internal working principle and working process of the system are studied. The auxiliary converter adopts the droop control method introducing virtual impedance to reduce the circulation effect of the parallel system. Combined with the battery pack charging characteristic curve, the constant current and constant voltage step-by-step charging management strategy is studied. The actual operating conditions of the system are simulated according to the auxiliary power supply system control logic. The system-level simulation on the MATLAB/Simulink platform shows that the output performance of each component is good, working in the rated parameters and meeting the working requirements of the auxiliary power supply system. Finally, the rationality, stability and robustness of the auxiliary power supply system model are proved by simulation and experimental comparison. This research provides a certain theoretical basis for the research of China standard EMU auxiliary power supply system, which has a certain significance.

**Keywords:** auxiliary power supply system; droop control; charging management strategy; system control logic

## 1. Introduction

High-speed EMU technology combines various complex technologies and the auxiliary power supply system is an important part of the EMU. It not only supplies power to all load devices outside the traction power system that need to be powered, but also ensures the normal operation of some parts of the EMU, which facilitates people's lives and travel. The normal operation of the auxiliary power supply system is a necessary condition for ensuring stable and safe operation of the EMU.

Development of auxiliary power supply system technology for EMUs in developed countries such as the United States, Germany, and Japan has been very mature. With the introduction and innovation of the EMU technology, the research on auxiliary power supply system technology in China has developed rapidly. The China Standard EMU is a newly developed EMU technology. Before that, China used more CRH (China Railway High-speed) EMUs. Various auxiliary power supply system designs have their own advantages and disadvantages. Therefore, domestic and foreign scholars have carried out more in-depth research on the EMU auxiliary power supply system.

Reference [1] mainly introduced the low voltage power supply system (LVPS) and its principle for a metro vehicle in an overseas project. Focusing on the efficiency of LVPS under the highest and the lowest limit input voltage. The efficiency characteristics of LVPS with the load changes were analyzed, which provides a reference for the design and selection of low-voltage power supply for

subway. A new control method of single-phase Phase-Locked Loop (PLL) function for the electrified railway converter system was presented in reference [2]. This method has been implemented on the converter of the auxiliary power supply system for the 8200 series electric locomotive in Korea. In reference [3], a composite control algorithm for the CRH2 EMU auxiliary power supply system was designed. The double closed-loop control algorithm based on current decoupling was used for the four-quadrant rectifier, while the double closed-loop control and repetitive control was designed for the three-phase voltage inverter. This control algorithm improved the stability and reliability of the auxiliary power supply system and its effectiveness has been verified through simulation and experiments. In reference [4], the loss model of the 1500 V power subway auxiliary inverter was established. The author estimated loss and efficiency of SiC MOSFET and Si IGBT systems under different switching frequencies. Analysis showed that the application of the SiC MOSFET could reduce the loss and improve the efficiency of the system. The SIC MOSFET was used to improve the topology of the auxiliary power supply system of the rail vehicle in reference [5,6]. It simplified the design of the converter, improved the working efficiency, reduced the switching loss and decreased the heat dissipation, benefiting the development of environmentally-friendly society. A simulation model of the EMU auxiliary converter system was established in reference [7]. It mainly introduced the implementation methods of HIL (Hardware in the loop) simulation based on FPGA (Field—Programmable Gate Array) simulation board. The simulation results verified the effectiveness of the model and proved the feasibility of the method. Different components of the auxiliary power supply system had been proposed and optimized in references [8–10], all methods improved the overall performance of the EMU auxiliary power supply system. In reference [11], the fault codes of the auxiliary power communication device of the 8200 series electric locomotive were studied, and the operational characteristics of the auxiliary power supply system were analyzed to provide a basis for the subsequent reliability research of the system. A flyback auxiliary power supply scheme was proposed and investigated with two transistors serially connected at the input [12]. This converter has an integrated transformer, and all of the series modules were operating synchronously, which was suitable for the applications with the high input voltage, multiple outputs and low-power consumption. In reference [13], the principle and influence of the circulating current of each series module were analyzed for the flyback converter with two series transistors and a control strategy was proposed to eliminate the influence of circulating current on the operation of the converter. A new type of EMU auxiliary power supply system topology was designed and a control strategy for improving the transmission efficiency of DC (Direct Current)-DC converter was proposed based on the new topology, which verified the load carrying capacity of the DC bus and the feasibility of the emergency power supply scheme by using the energy storage device [14]. Reference [15] proposed a novel control strategy of the auxiliary converter parallel system to avoid coupling interference of output voltage. A scheme was presented in reference [16] that the uninterrupted power supply of the auxiliary power supply system could be realized by optimizing the control strategy. In reference [17], a two-stage cascaded auxiliary power supply for MMCs (Modular Multi-Level Converter) was proposed with 300 V–3000 V input and 24 V output, and the design scheme for auxiliary power supply was given. New control strategies and modulation techniques were respectively presented in reference [18], which improved the quality of the output voltage of the auxiliary power supply system and the feasibility of the new strategies was justified by experiments. In reference [19], taking the EMU auxiliary power supply performance as the standard, the evaluation and optimal solution of the comprehensive performance were obtained by judging the calculation weight and each specific index via matrix. In reference [20], the influences of different starting modes of large-capacity and multi-loading of EMU auxiliary power supply system on the overall system were compared through simulation, and a method of step-by-step starting of load was presented and ensured the safe operation of the auxiliary power supply system.

At present, most of the research only studied individual components of the EMU, and the corresponding simulation and optimization had been carried out; however, it lacked systematic research for the EMU. For example, the references [5–10] only studied the auxiliary converter. The reference [11]

established a new auxiliary power supply system model; however, research and analysis on existing models had not been conducted, and the feasibility of the model had not been verified. Two new optimization methods were proposed in references [19,20]; however, it lacked the verification of the real vehicle experiment. As a new type of vehicle, China Standard EMU has not seen any literature on the overall research of auxiliary power supply system. Therefore, a systematic simulation study of the auxiliary power supply system of China standard EMU has filled the gap of the theoretical and practical research. It provides a certain theoretical basis for the follow-up study of China standard EMU and has certain practical value.

Firstly, the single model such as auxiliary converter, charger and single-phase inverter is built, and then the overall model of the auxiliary power supply system of China standard EMU is built in this paper. The droop control method is used to reduce the influence of circulation in the auxiliary converter parallel system. The control algorithm is used to realize the constant current and constant voltage charging strategy of the battery pack. The operation process of the system under different working conditions are simulated in the MATLAB/Simulink according to the auxiliary power supply system control logic. Finally, the rationality, stability and robustness of the system are verified by the simulation, and the comparison has been performed with the real vehicle experiments.

## 2. EMU Auxiliary Power Supply System

China Standard EMU is composed of 8 wagon groups, including 4 moving and 4 towing groups, or 2 sets of traction units. The auxiliary power supply system is mainly composed of an auxiliary power supply and its utilization equipment. The auxiliary power supply system is composed of auxiliary converters, chargers, battery packs, etc. A simplified schematic diagram of the EMU auxiliary power supply system is shown in Figure 1.

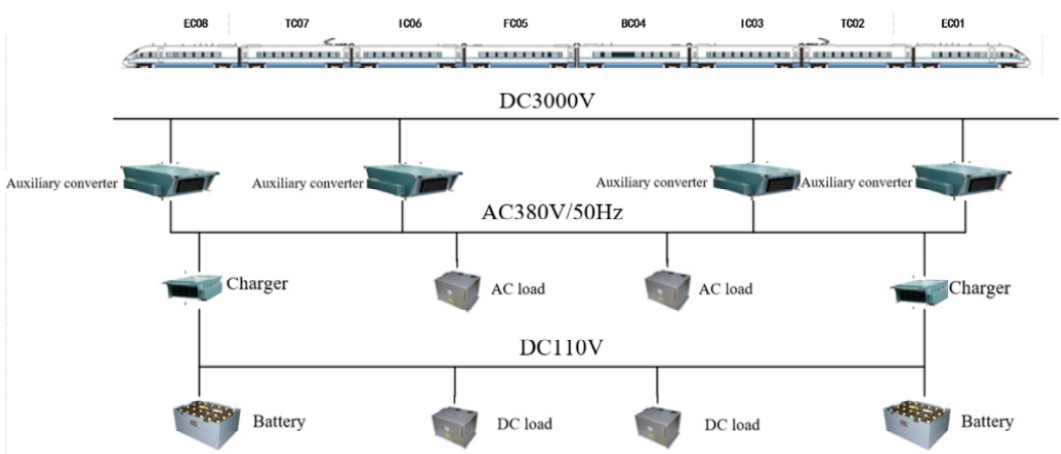

**Figure 1.** Simplified schematic diagram of the EMU auxiliary power supply system.

The auxiliary power supply system uses the main line power supply mode to supply power for all the electric equipment except the traction system on the EMU. The electric power enters the auxiliary converter through the secondary winding of the traction transformer or the DC link of the main converter through the pantograph, and the auxiliary converter is processed to output 380 V/50 Hz three-phase alternating current to the AC (Alternating Current) power supply bus. The charger obtains electric energy from the AC bus, converts the three-phase AC output voltage of the auxiliary inverter into 110 V DC voltage, and charges the EMU battery pack. The battery pack also provides emergency power for the auxiliary power supply equipment without interruption of the power failure. The EMU load is divided into AC load and DC load. The AC load mainly includes AC equipment such as charger, main transformer cooling fan and air conditioner. They all take power from AC 380 V power bus. DC load mainly includes battery pack, single-phase inverter and electric lamp. DC equipment is taken from the DC 110 V power bus.

The working control logic of the EMU auxiliary power supply system is as follows: firstly, the bus voltage of the 3000 V DC link is started, and then an auxiliary converter (main auxiliary) is started. Secondly other auxiliary converters are paralleled. After the parallel system of the auxiliary converter is connected stably, the charger starts the work after obtaining stable three-phase AC power from the auxiliary converter. The charger supplies power to the battery pack and the DC load, and an overall stable operation of the auxiliary power supply system is achieved.

## 3. Research on Key Problems of Auxiliary Power Supply System

The key modules of the EMU auxiliary power supply system mainly include auxiliary converters, chargers, batteries and related loads. The topology of the auxiliary power supply system is shown in Figure 2.

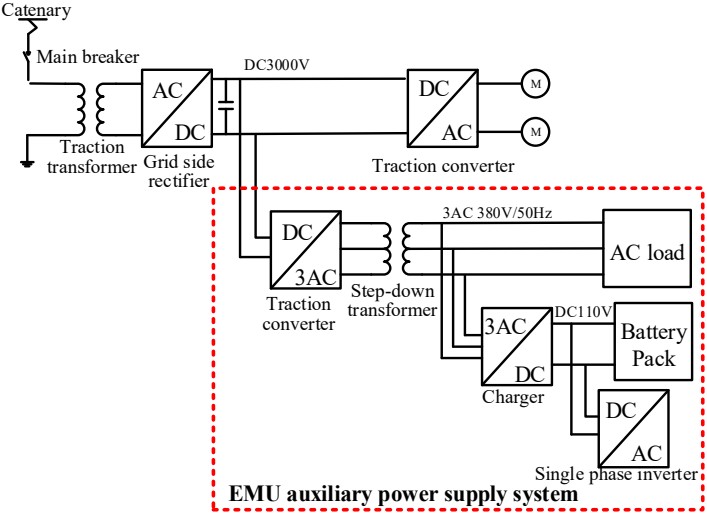

**Figure 2.** Topology diagram of the EMU auxiliary power supply system.

At present, the following key issues exist in the simulation research of auxiliary power supply systems:

(1) In the overall actual operation of the auxiliary power supply system, the instantaneous values of the output voltages are different due to the different starting times of the two inverters, resulting in a circulation between the two converters. This affects the normal operation of the converter and even causes the converter to be paralyzed. Therefore, it is necessary to adopt a certain control strategy to minimize the circulation between the two inverters.

(2) In the auxiliary power supply system, the charging mode of the battery pack is based on the *constant current/constant voltage* control strategy, and it is quite complicated. In order to simulate the real working conditions of the whole system, the output voltage and current of the charger need to be constantly changed according to the State of Charge (SOC) of the battery pack. Therefore, it is necessary to study the control strategy of charging the battery pack.

(3) The entire auxiliary power supply system with a large number of components is in a large scale, and the operation of the system is controlled by certain logic. Therefore, research on the control unit of the system is required.

In view of the above key problems, the topological structure and control principle of the key modules of the auxiliary power supply system are studied. The following research is made on the auxiliary converter parallel system, the battery pack charge management strategy and the overall control logic of the system.

### 3.1. The Stable Operation of Auxiliary Converter Parallel System

A single auxiliary converter converts the 3000 V DC voltage of the traction converter into stable three-phase 380 V/50 Hz AC voltages, which can quickly and stably respond to load abrupt conditions. The basic topology is shown in Figure 3.

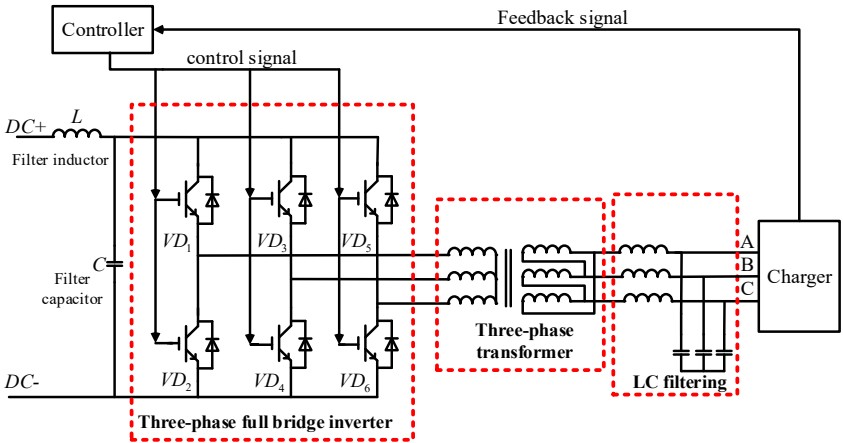

**Figure 3.** Basic topology of China Standard EMU auxiliary converter.

The auxiliary converter control adopts the SVPWM (Space Vector Pulse Width Modulation) control method, and the SVPWM modulation pulses drive the auxiliary converter to output a stable voltage. Figure 4 shows the feedforward dual-loop control block diagram with the decoupling of output voltage and current. The inverter output voltage $u_o$, the output current $i_o$, and the bridge arm current $i_L$ are transformed into a synchronous rotation dq coordinate system. The inverter output voltage d-axis component $u_{od}$ tracks the d-axis component reference value $u_{odref}$ through the PI (Proportion Integral) controller. PI controller output value is superimposed on the output current d-axis component $i_{od}$ and feedforward output voltage q-axis compensation component $-wcu_{oq}$ as the d-axis component reference value $i_{Ld}^*$, for the bridge arm current $i_{Ld}$. The bridge arm current $i_{Ld}$ tracks the d-axis component reference value $i_{Ld}^*$ through the PI regulator. The PI controller output value is superimposed on the output voltage d-axis component $u_{od}$ and the feedforward bridge arm current q-axis compensation component $-wLi_{Lq}$ as the d-axis component of bridge arm voltage $u_d$. Similarly, the q-axis component of the bridge arm voltage $u_q$ can be obtained. Convert the d and q components of the bridge arm voltage into three-phase modulated waves $u_m$ (m = a,b,c) under three-phase stationary coordinates. $u_m$ generates a 6-way switch drive signal through the modulation module. This control method with a strong robustness can stabilize the output voltage of the auxiliary power supply system, ensuring the system will quickly return to a steady state when the load is switched.

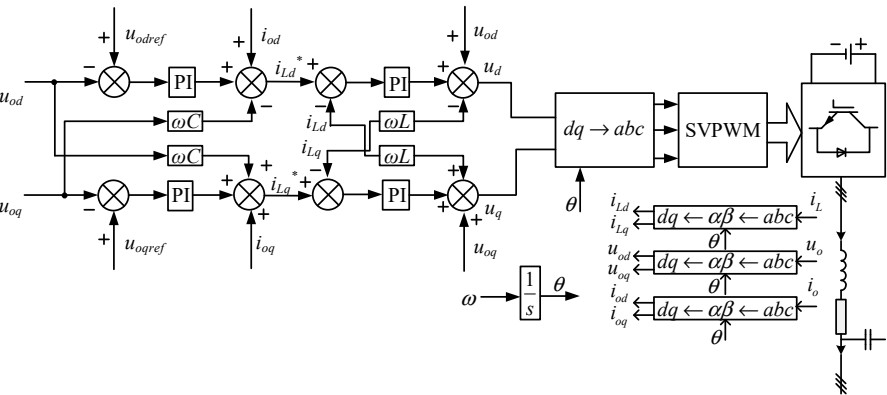

**Figure 4.** Feedforward dual-loop control block diagram with the decoupling of output voltage and current.

The auxiliary converter uses parallel operation in actual work. The basic simplified model of two auxiliary converters in parallel is shown in Figure 5, where $E_1 \angle \varphi_1$ and $\dot{I}_1$ are respectively the no-load output voltage and the output current of the auxiliary converter 1. $r_1 + jX_1$ is the output impedance of the auxiliary converter 1, equivalent to the sum of the output impedance of the auxiliary converter 1 and the line impedance value to the AC bus. The parameters of the two auxiliary converters are the same. $Z_L$ is the load and $V \angle 0$ is the load voltage. $\dot{I}_H$ shown in Figure 5 is system current circulation.

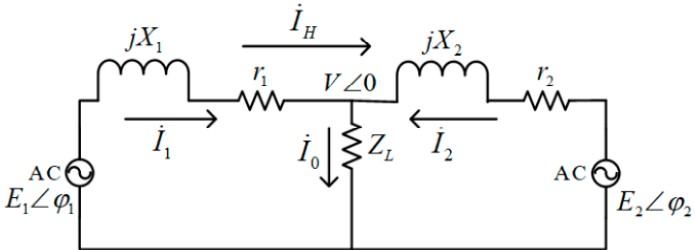

**Figure 5.** Simplified model diagram of auxiliary converter parallel system.

In terms of the auxiliary converter parallel system, the parameters of the two auxiliary converters are the same, which is $r_1 + jX_1 = r_2 + jX_2 = Z$, so the following formula (1) can be derived:

$$\dot{I}_H = \frac{\dot{I}_1 - \dot{I}_2}{2} = \frac{1}{2}\left(\frac{\dot{E}_1 - \dot{V}}{r_1 + jX_1} - \frac{\dot{E}_2 - \dot{V}}{r_2 + jX_2}\right) = \frac{\dot{E}_1 - \dot{E}_2}{2Z} \tag{1}$$

where $\dot{I}_1$, $\dot{I}_2$ and $\dot{I}_H$ are vector form of two output currents and the system loop current, respectively.

It can be seen from the above equation that the system circulating current value is proportional to the no-load output voltage difference of the two auxiliary converters, and inversely proportional to the equivalent output impedance of the single inverter.

Larger circulations can affect the normal operation of the converter and even cause paralysis of the system. The common way to reduce the circulation is the droop control method introducing virtual impedance [21]. The structure diagram is shown in Figure 6. It mainly includes power measurement module, droop characteristic control module, virtual impedance module and current and voltage double closed loop control module.

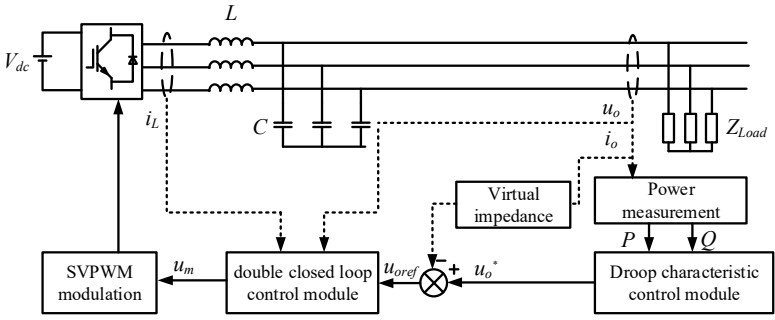

**Figure 6.** Structure diagram of the droop control method introducing virtual impedance.

According to the instantaneous power calculation method, when the coordinate system is a two-phase dq rotating coordinate system, the calculation formula of the instantaneous active power $p$ and the reactive power $q$ are:

$$\begin{cases} p = \dfrac{3}{2}\left(u_{od}i_{od} + u_{oq}i_{oq}\right) \\ q = \dfrac{3}{2}\left(u_{oq}i_{od} - u_{od}i_{oq}\right) \end{cases} \tag{2}$$

The inverter output active power $p$ and reactive power $q$ obtained by Formula (2) have an AC component and a relatively high frequency. Therefore, a low-pass filter is added to obtain the average active power $P$ and the average reactive power $Q$. As shown in Formula (3), $\omega_c$ is the cutoff frequency of the low pass filter. $P$ and $Q$ are passed into the droop feature module as the final power input semaphore.

$$\begin{cases} P = \frac{\omega_c}{s+\omega_c}p \\ Q = \frac{\omega_c}{s+\omega_c}q \end{cases} \tag{3}$$

The droop characteristic control module is mainly realized by the absorption of active power and reactive power, which are triggered by the difference of the amplitude and phase of the output voltage of each single module in the parallel system. In the parallel system, the output resistive components of the auxiliary converter is small, and the difference between the single output voltage and the bus voltage is small, so it supposes: $X_1 = X_2 = X$, $r_1 = r_2 \approx 0$, $sin\varphi_i = \varphi_i$, $cos\varphi_i = 1$ $(i = 1,2)$. Then the output active power $P_1$ and reactive power $Q_1$ of the auxiliary converter 1 are:

$$\begin{cases} P_1 \approx \frac{U_1 V}{X}\varphi_1 \\ Q_1 \approx \frac{U_1 V - V^2}{X} \end{cases} \tag{4}$$

where $U_1$, $V$ and $\varphi_1$ are output voltage amplitude of the auxiliary converter 1, common bus voltage amplitude and the difference between the phase angle of the voltage at the outlet of the converter 1 and the phase angle of the bus voltage, respectively.

The output active and reactive power $P_2$ and $Q_2$ of the auxiliary converter 2 shares the resemblance:

$$\begin{cases} \Delta P = P_1 - P_2 = \frac{U_1 V}{X}(\varphi_1 - \varphi_2) = \frac{U_1 V}{X}\Delta\varphi \\ \Delta Q = Q_1 - Q_2 = \frac{U_1 - U_2}{X}V = \frac{\Delta U}{X}V \end{cases} \tag{5}$$

It can be seen from the above formulas that the change of the output active power is mainly affected by the phase of the output voltage, and the output reactive power is mainly affected by the change of the output voltage amplitude. The basic governing equation for the drooping characteristic can be obtained as follows:

$$\begin{cases} f = f_0 - K_p P \\ U = U_0 - K_q Q \end{cases} \tag{6}$$

where $f$ and $U$ are frequency and amplitude reference values of the auxiliary converter output voltage, $f_0$ and $U_0$ are the frequency and amplitude of the output voltage when the auxiliary converter is unloaded, $K_P$ and $K_q$ are droop parameters of active and reactive power, respectively.

Figure 7 shows the drooping characteristic curve of the parallel system. The droop control can achieve the active power and reactive power balance between modules.

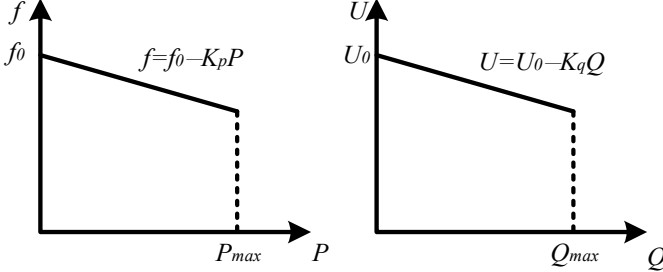

**Figure 7.** Parallel system droop characteristic curve.

According to the *P* and *Q* calculated by the power measurement module, the output voltage amplitude $u_0{}^*$ by the inverter can be obtained by the droop control. Since the output side of the inverter is equipped with a filter inductor under actual working conditions, the output impedance of the inverter is biased when the filter inductance is large, that is, the equivalent output impedance is biased at the power frequency. Therefore, the virtual impedance $Z_{vir}$ is designed to be inductive. After passing through the virtual impedance module, a new voltage reference value $u^*{}_{oref}$ is obtained. Considering the influence of the virtual impedance, the calculation formula of $u^*{}_{oref}$ in the dq coordinate system is shown in the Formula (7).

$$\begin{cases} u^*{}_{odref} = u_{od}{}^* + \omega L\text{vir}I_{oq} \\ u^*{}_{oqref} = u_{oq}{}^* - \omega L\text{vir}I_{oq} \end{cases} \tag{7}$$

where $u^*{}_{odref}$ and $u^*{}_{oqref}$ are the dq components of the output voltage reference value $u^*{}_{oref}$ after passing through the virtual impedance module, $u_{od}{}^*$ and $u_{oq}{}^*$ are the dq components of the output voltage reference value $u_o{}^*$ after the droop control module, $\omega$ and $L_{vir}$ are the angular frequency in the dq coordinate axis and the virtual inductance, respectively.

Finally, the reference voltage $u^*{}_{oref}$ is generated by the feedforward dual-loop decoupling control of the voltage and current, which is shown in Figure 4 to generate the trigger $u_m$ of the inverter, and the $u_m$ drive SVPWM control generates a trigger pulse to control the output voltage of the inverter.

In the auxiliary converter parallel system, the droop control method with virtual impedance is used to reduce the system circulating current value, achieve the balance between active power and reactive power, and improve the quality of the output voltage of the parallel system.

### 3.2. Battery Pack Constant Current Constant Voltage Charging Design

As the auxiliary stage of the auxiliary converter, the charger mainly supplies a sTable 110V DC power to the battery pack and DC load. The basic topology of the China standard EMU charger is shown in Figure 8.

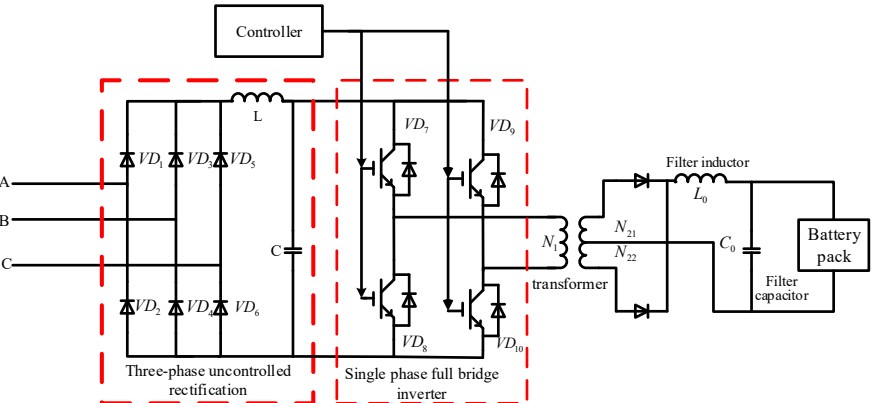

**Figure 8.** Basic topology of China Standard EMU charger.

The charger also uses the current and voltage double closed loop control strategy. The output current and voltage values are compared with the reference value and then controlled by PI method. The output signal is sent to the IGBT pulse control device to regulate the stable output of the voltage and current.

Using the lithium battery as an example for the China standard EMU auxiliary power supply system battery pack, there are various polarization phenomena during the charging process. The ideal state of battery charging is to reach the goal of short charging time and small battery damage [22]. Increasing the charging current can shorten the charging time, however, it also causes irreparable damage to the battery. Considering three factors of charging time, battery capacity and life, American

scientist Joseph A. Mas has proposed a rapid battery charging curve through a large number of experimental studies, which is shown in Figure 9.

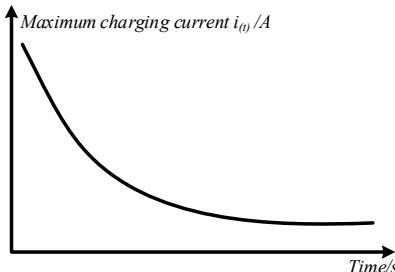

**Figure 9.** Fast charging curve.

The battery pack charging diagram shown in Figure 10 can be obtained by considering the battery pack charging requirements, the fast charging curve and the actual charging conditions of the battery pack. The charging current is expressed by the charging magnification C. There are two stages in charging the battery pack. The first stage is the five-step constant current stage, the battery pack is first charged quickly with the maximum current (0.5 C) that can be accepted. When the battery pack voltage rises to the set voltage $V_1$, the charging current is reduced to 0.25 C. When the charging current decreases, the battery voltage decreases by a certain magnitude, and then continues to rise at a lower constant charging current until it reaches $V_1$ again, and this process is repeated. The second stage is the constant voltage phase, the purpose is to keep the battery pack voltage at the set voltage $V_1$ until the end of charging.

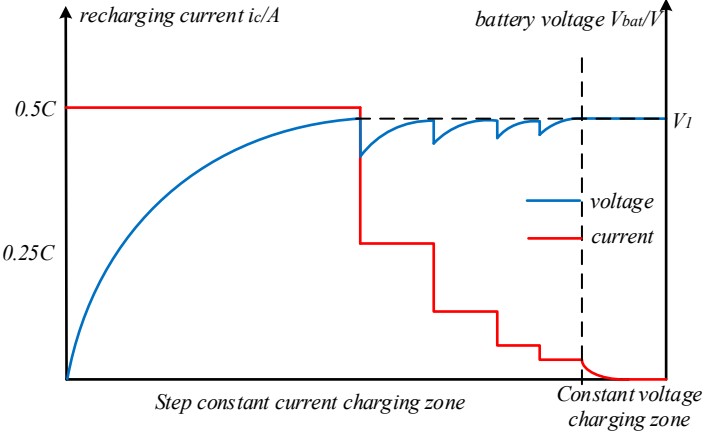

**Figure 10.** Battery pack charging diagram.

The battery pack energy management system (BMS) sends the maximum allowable charging current to the charger in real time, and the charger controls the output voltage and current according to the battery state. The basic parameters of the Chinese standard EMU battery pack are shown in Table 1. According to above analysis, a battery pack charging flow chart can be obtained, as shown in Figure 11.

**Table 1.** Parameters of China Standard EMU battery pack.

| Parameter | Value |
|---|---|
| rated voltage | 103.5 V |
| Maximum voltage | 126 V |
| capacity | 190 Ah |
| set voltage $V_1$ | 117 V |

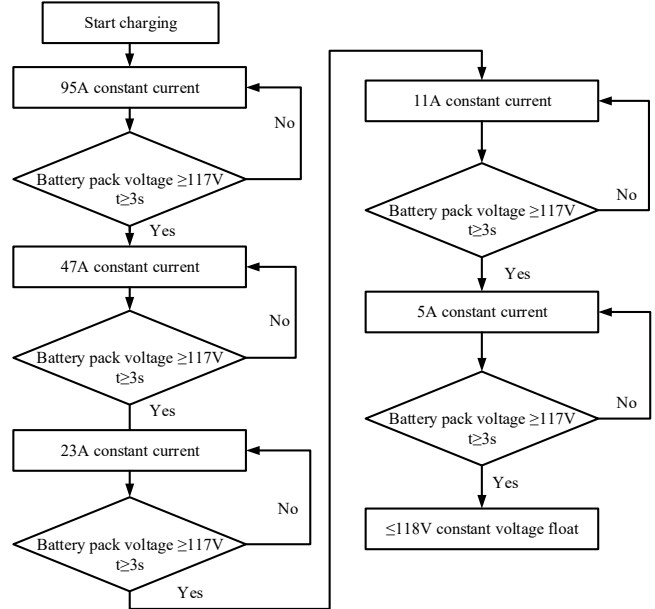

**Figure 11.** Standard EMU battery pack charging flow chart.

### 3.3. Design of Control Unit for Auxiliary Power Supply System

The working process of the auxiliary power supply system includes the starting process and the steady state running process. The auxiliary power supply system control unit provides logic control for the system start-up.

It can be seen from Figure 1 that the China standard EMU is a symmetrical structure. Combined with the starting conditions of the China standard EMU, the start-up sequence is shown in Table 2 by taking two auxiliary converters, one charger, one battery pack and several loads as an example.

**Table 2.** Logic start-up sequence of auxiliary power supply system.

| Time (s) | Starting Part |
| --- | --- |
| 0 | Auxiliary converter 1 |
| 0.2 | Auxiliary converter 2 |
| 0.4 | Charger |
| 0.8 | Single-phase inverter |

## 4. Modeling Simulation Analysis and Experimental Verification

The overall connection diagram of the auxiliary power supply system is shown in Figure 12. According to Figure 12, the simulation model of the Chinese standard EMU auxiliary power supply system is built on the MATLAB/Simulink platform, which is shown in Figure 13. The simulation parameters are shown in Table 3.

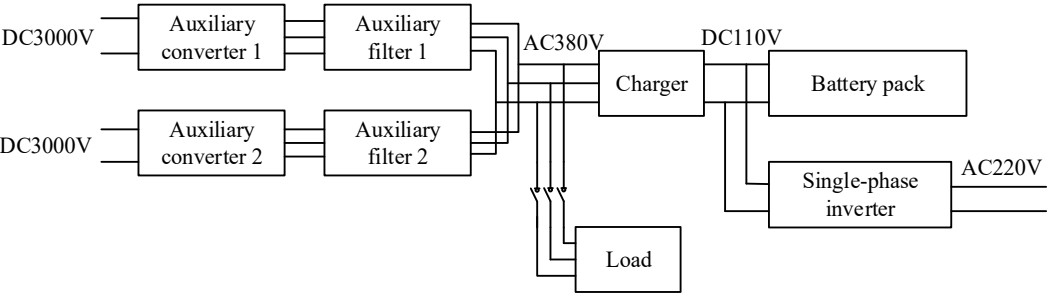

**Figure 12.** Diagram of the overall connection of the auxiliary power supply system.

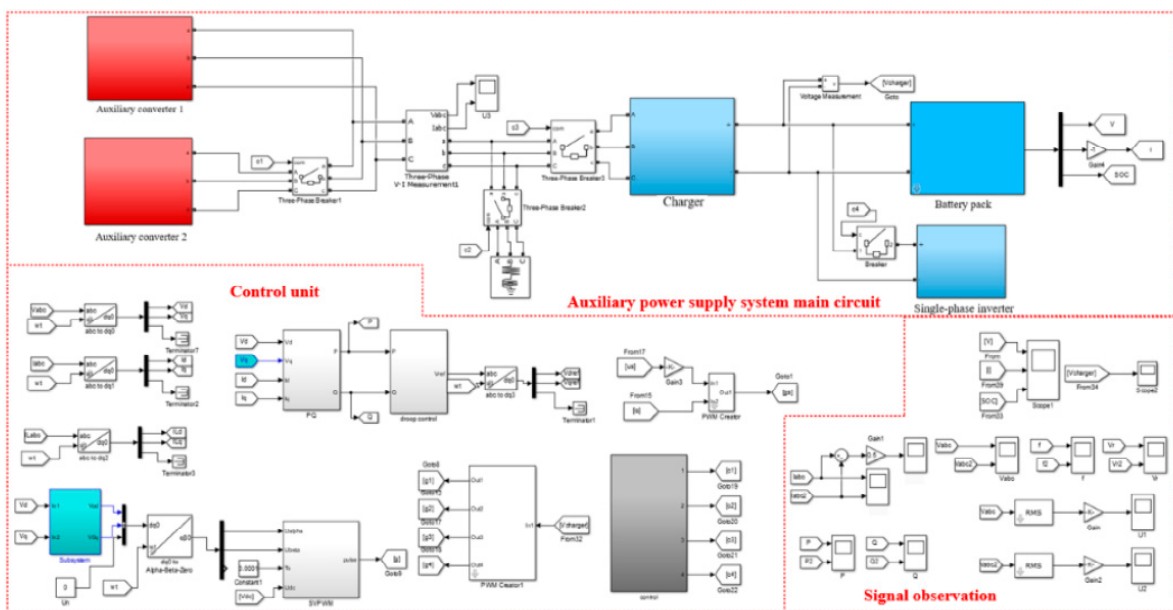

**Figure 13.** Power supply system simulation model.

**Table 3.** Power supply system simulation parameters.

| Parameter | Value |
| --- | --- |
| Input DC voltage $V_{in}$ | 3000 V |
| Auxiliary converter filter inductor/capacitor | 0.35 mH/50 μF |
| Charger one side inductance / capacitance | 2 mH/5000 μF |
| Charger two side inductance / capacitor | 0.5 mH/3000 μF |
| Transformer ratio | 400:116 |
| Switch sampling frequency $f_s$ | 10 kHz |
| Auxiliary converter rated power $P_1$ | 200 kW |
| charger rated power $P_2$ | 60 kW |

### 4.1. Auxiliary Power Supply System Startup Simulation Analysis

The whole system is started according to the control logic of Table 2. After the two auxiliary converters are connected in parallel, the AC voltage is supplied to the after-stage charger. The charger is powered after the charger is started. The simulation of the auxiliary converter, charger and single-phase inverter can be obtained by simulation, as shown in Figure 14. Figure 15 is the Circulation comparison and THD (Total Harmonic Distortion) value of auxiliary converter parallel system.

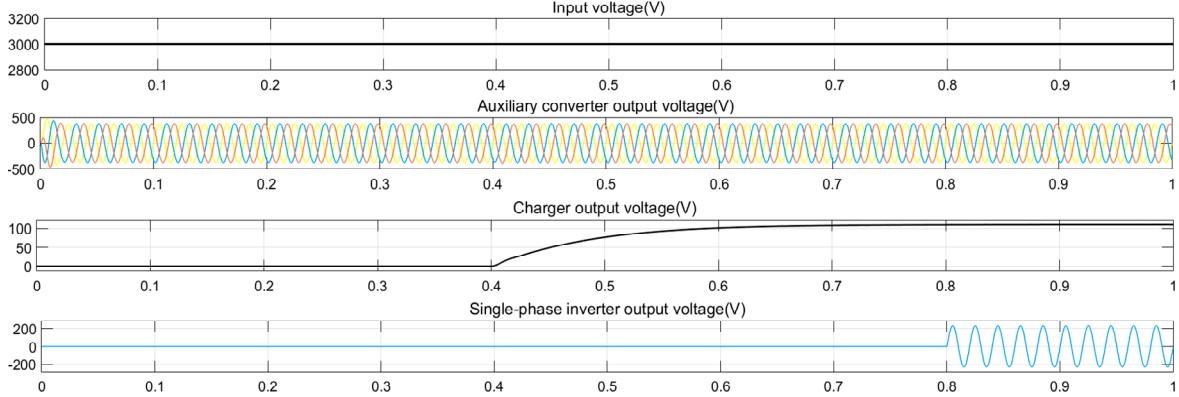

**Figure 14.** Power supply system step-by-step simulation results.

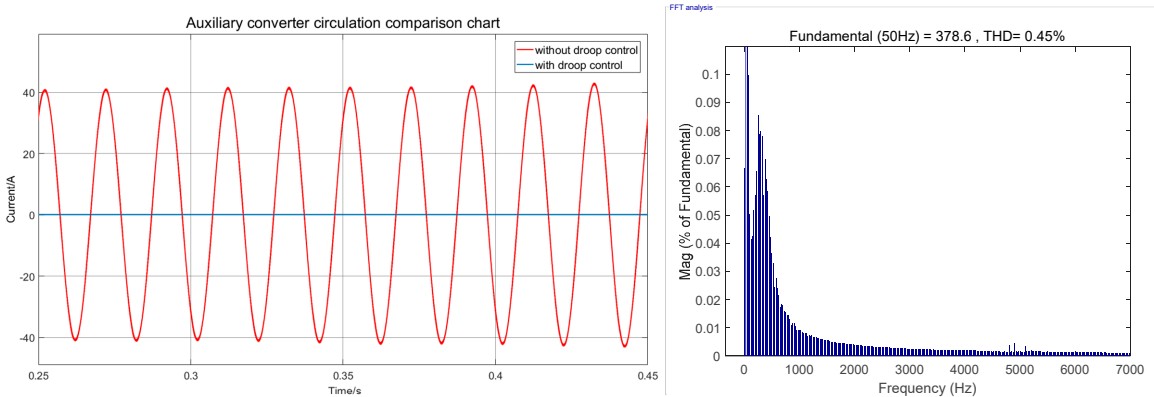

**Figure 15.** Comparison and THD value of auxiliary converter parallel system.

It can be seen from the above waveform that the auxiliary converter system has almost no circulation effect after the parallel connection. The output voltage of the auxiliary converter is stable at the value of 378.6 V and the THD value is 0.45%. The output voltage response of the charger is fast, and the DC voltage output of 109.8 V meets the fluctuation requirement of ±5% for the 110 V DC voltage. The Single-phase inverter stabilizes the 219.2 V AC output voltage. The whole system meets the requirements of actual working parameters.

Controlling the output current and voltage of the charger according to the designed battery pack charging management strategy, the battery pack charging simulation waveform can be obtained, which is shown in Figure 16. The initial SOC of battery pack is 10%, with an initial voltage of 104 V.

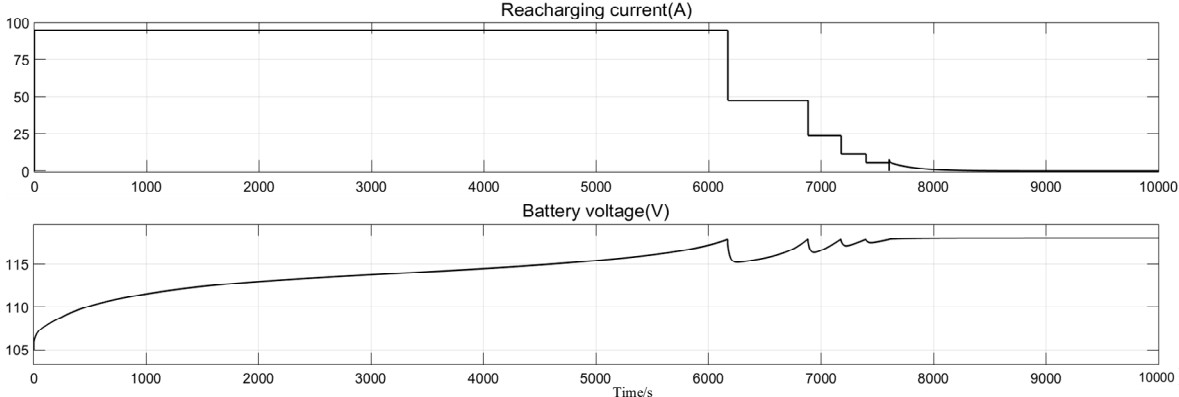

**Figure 16.** Pack charging simulation results.

It can be seen from the battery pack charging simulation waveform that the simulation results are in accordance with the charging management strategy shown in Figures 10 and 11. Since the initial voltage of the battery pack is less than 117 V, the charger first outputs a large current of 0.5 C (95 A) for constant current charging. When the battery reaches the set voltage of 117 V for the first time, the step constant current charging is started. After multiple steps of constant current charging, the charger performs constant voltage charging on the battery to maintain the battery voltage at 117 V. After charging is completed, the battery voltage reaches 117.3 V.

*4.2. Simulation Analysis of Auxiliary Power Supply System Stability*

In order to verify the stability and robustness of the operation of the auxiliary power supply system, the output AC bus load switching simulation analysis is carried out in MATLAB. After the parallel output of the auxiliary converter is stabilized, the system operates at the rated load state. When the load is suddenly cut at 2.2 s and 2.4 s, the system works at 60% load state and 10% load state, respectively. When the load is suddenly increased at 2.6 s and 2.8 s, the system works at 60% load state

and rated load state respectively. Figure 17 shows simulation results of the auxiliary converter output voltage and current. The current transits fast and there is no obvious oscillation.

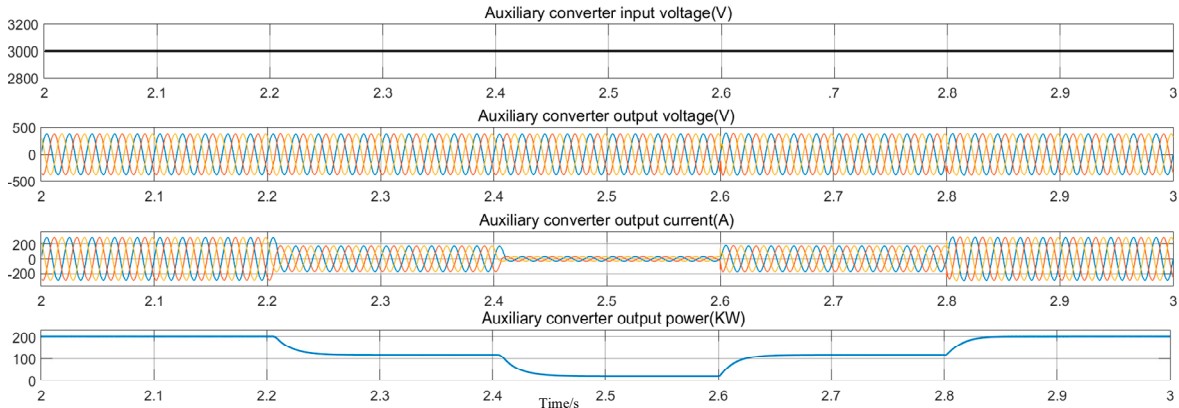

**Figure 17.** AC bus load switching simulation results.

The operating status of the auxiliary power supply system with rated load has been simulated, and the rated load values of each monomer model are shown in Table 4. It can be seen from the simulation waveform diagram in Figure 18 that the voltage output of each part is stable and meets the actual working requirements.

**Table 4.** Auxiliary power supply system rated load value.

| Component | Rated Load Valueb (kW) |
| --- | --- |
| Auxiliary converter | 200 |
| Charger | 60 |
| Single-phase inverter | 3.5 |

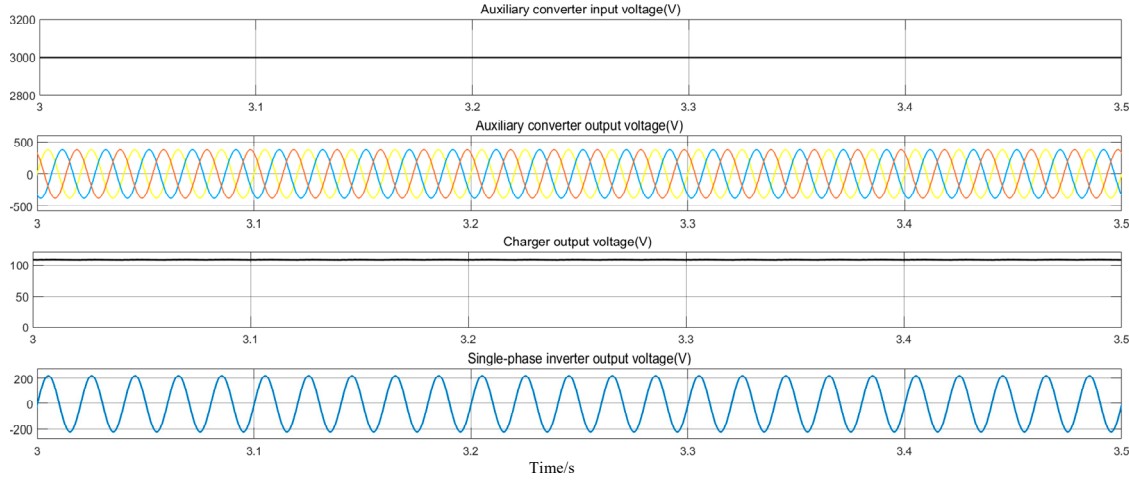

**Figure 18.** Simulation results under rated load conditions.

### 4.3. Experimental Verification

The auxiliary power supply system experiment was carried out on the experimental platform of China Standard EMU, and the output waveform of the auxiliary power supply system under different working conditions was measured and compared with the simulated waveform. The experimental platform of China Standard EMU auxiliary power supply system is shown in Figure 19. The main components are: auxiliary converter, charger and vehicle single-phase inverter. The system is bulky and was placed in the open air, and the control section was placed indoors.

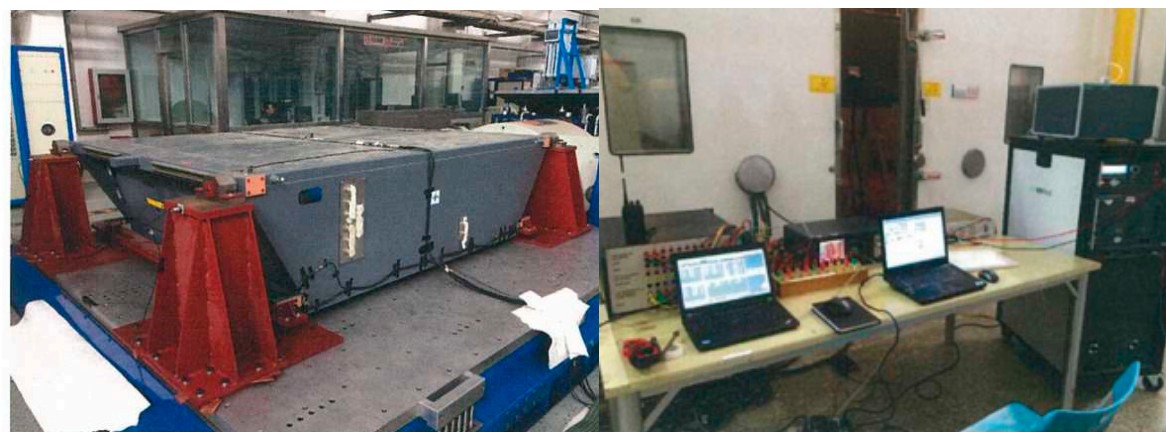

**Figure 19.** Standard EMU Experimental Platform.

Measured output waveform of output waveform of the battery charging experiment. In order to better verify the performance of the whole system, the initial set voltage of the battery pack is set to 104 V, and the experimental waveform is shown in Figure 20. Through comparison and analysis with Figure 16, it can be seen that the battery pack adopts the charging strategy of constant current and constant voltage, and the charging simulation results is the same as the experimental results.

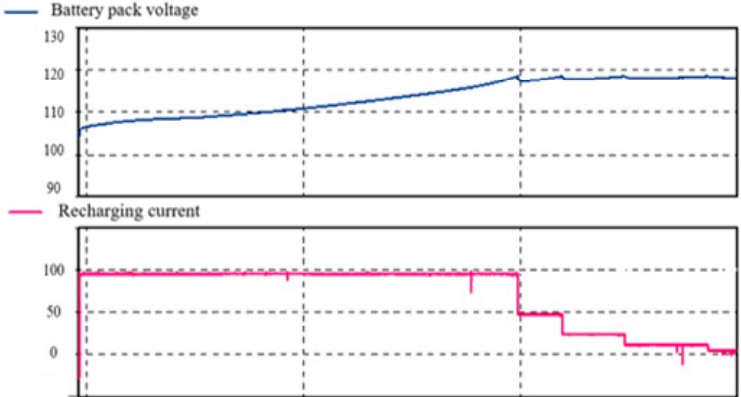

**Figure 20.** Pack charging experiment results.

Measured output waveform of the AC bus when load switching, the experimental conditions are consistent with the simulation conditions. The experimental waveform is shown in Figure 21. After comparing with Figure 17, it can be seen that the simulation results are basically consistent with the experimental results for the auxiliary converter output voltage and current during the load switching process. Table 5 compares the simulated and experimental data of the auxiliary converter output current under different operating conditions.

**Table 5.** Converter output current simulation and experimental data comparison.

| Detection Part | Simulation | Experiment |
| --- | --- | --- |
| Auxiliary converter current at 10% load | 36.5 A | 30.44 A |
| Auxiliary converter current at 60% load | 185.6 A | 178.53 A |
| Auxiliary converter current at 100% load | 302.2 A | 296.77 A |

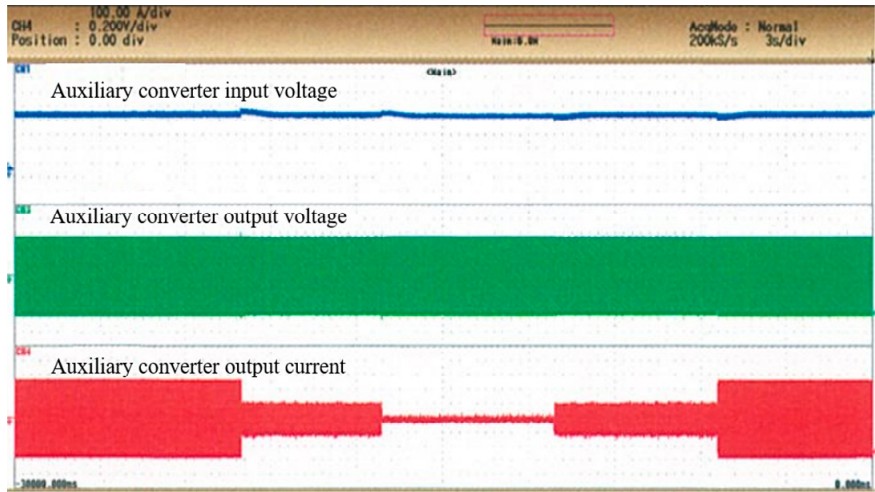

**Figure 21.** Bus load switching test results.

Measured output waveform of the auxiliary power supply system under rated load conditions is shown in Figure 22. Figure 22 is compared with the simulated waveform shown in Figure 18. Table 6 shows that the difference between the simulated data and the measured data is small, indicating the feasibility and rationality of the overall simulation of the auxiliary power supply system.

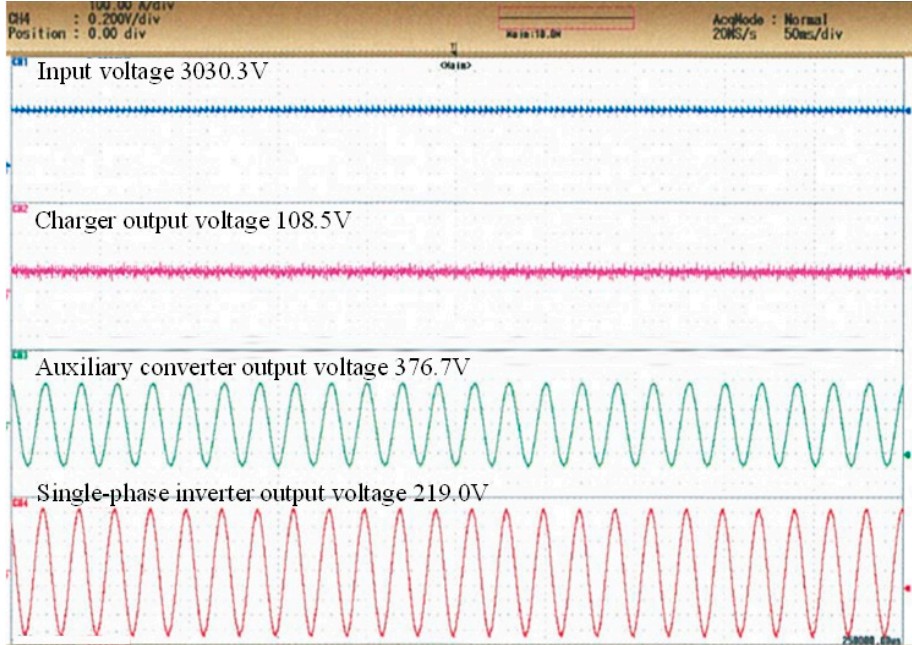

**Figure 22.** Results under rated load conditions.

**Table 6.** Difference between simulation and experimental data under rated load.

| Detection Part | Simulation | Experiment |
|---|---|---|
| Input voltage | 3000 V | 3030.3 V |
| Auxiliary converter voltage | 3AC378.6 V | 3AC376.7 V |
| Charger voltage | DC109.8 V | DC108.5 V |
| Single-phase inverter voltage | AC219.2V | AC219.0V |

## 5. Conclusions

This paper conducts simulation experiments on the auxiliary power supply system of China Standard EMU. The actual operating conditions of the auxiliary power supply system has been simulated according to the corresponding control logic. It is verified that the auxiliary power supply system simulation model has good stability and global robustness. Through comparative analysis, the following conclusions can be drawn:

(1) According to the auxiliary power supply system control unit, the system is started step by step, and there is no large shock during the start-up process. Two auxiliary converters have good parallel output performance and each component reaches the working rated parameters. The simulation model meets the requirements of the auxiliary power supply system.

(2) Combined with the battery charging characteristic curve, the two charging stages of five-step constant current charging and constant voltage charging are simulated. Battery charging simulation curve meets charging management strategy requirements.

(3) The auxiliary power supply system has stable output in all parts under different working conditions. The output ripple is small during the load switching process and the overall anti-interference ability is strong.

(4) In the test platform of China standard EMU auxiliary power supply system, the output of the system under different working conditions is measured. In the comparative analysis it can be seen that the simulation data is basically fitted with the actual data, indicating that the auxiliary power supply system model has practical reference value.

In this paper, the component-level simulation is upgraded to the system-level simulation, and the overall simulation study of the Chinese standard EMU auxiliary power supply system is carried out. According to the real vehicle topology, the auxiliary converter, the charger and other single models are built, and the overall system is built according to the real vehicle logic control structure. Furthermore, it studies the interior working principle and working prices of the system. The real vehicle experiment was carried out on the Chinese standard EMU experimental platform. The rationality, stability and robustness of the auxiliary power supply system model are proved by comparative analysis. The above research fills the blank of the overall research of China standard EMU auxiliary power supply system, providing a certain theoretical basis for the research of China standard EMU auxiliary power supply system. It also provides a certain reference value for the future maintenance of China Standard EMU. Moreover, future generations can perform deeper research based on this paper; for example, dragging the China standard EMU problems diagnosis into the simulation system. In the meantime, it also provides a new idea for the follow-up study of other vehicles, which is of great significance.

**Author Contributions:** Conceptualization, L.Y. and M.S.; Methodology, L.Z.; Validation, L.Y. and B.P.; Formal Analysis, M.S.; Writing—Original Draft Preparation, L.Y. and B.P.; Writing—Review & Editing, L.Z.; Supervision, L.Z.

**Funding:** This research was funded by the National Key R&D Program of China, grant number 2016YFB-1200601-B22.

**Conflicts of Interest:** The authors declare no conflict of interest. The funders had no role in the design of the study; in the collection, analyses, or interpretation of data; in the writing of the manuscript, or in the decision to publish the results.

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
