# Peer review of "Simulation Research on Auxiliary Power Supply System of China Standard EMU"

_electronics, doi:10.3390/electronics8060647_

Round 1
Reviewer 1 Report
This work conducts simulation experiments on the auxiliary power supply system of China 311 Standard EMU. The presented idea is interesting, and the simulations validate the effectiveness of the proposed scheme. However, there are some issues to be improved:
1- For the first time, the authors must define abbreviations such as PQ, CRH and ....
2- As there are a large number of variables, parameters, and sets, the authors should provide nomenclatures in the first section of the paper. It helps the reader to follow the paper conveniently.
3- The contributions of the paper are not clear.
4- In the introduction section, the literature review should be expanded. Moreover, the references are old. Please review the below references in the introduction section:
-- Investigation and implementation of an input-series auxiliary power supply scheme for high-input-voltage low-power applications. IEEE Transactions on Power Electronics. 2018 Jan;33(1):437-47.
-- Robust and Fast Voltage-Source-Converter (VSC) Control for Naval Shipboard Microgrids. IEEE Transactions on Power Electronics. 2019 Jan 30. DOI: 10.1109/TPEL.2019.2896244
-- The Research on Efficiency of Metro Vehicle Low Voltage Power Supply Under Limit Input Voltage. In2018 3rd International Conference on Smart City and Systems Engineering (ICSCSE) 2018 Dec 29 (pp. 275-278). IEEE.
-- Secondary load frequency control of time-delay stand-alone microgrids with electric vehicles. IEEE Transactions on Industrial Electronics. 2018 Sep;65(9):7416-22.
5- Please highlight the contributions of the paper in the last paragraph of the introduction section.
6- What's the computation effort of the proposed method compared to other references?
7- The control signals for each simulation are NOT given. This makes it more difficult to understand the benefit of the proposed approach. The author must give the control signals obtained for simulations.
8- It is strongly recommended that the authors check English grammar and vocabulary. Professional English editing service might be a good choice.
Author Response
Dear reviewer:
I appreciate your attention on and response to this paper. Please review the following responses to referee comments in the file 1.
Thank you very much for all your help and looking forward to hearing from you soon.

Reviewer 2 Report
The authors mainly present a simulation study of an auxiliary power supply system in a Chinese standard EMU. The paper describes the modules it is composed of and the control that is performed in each one. Therefore, there is no study of control methods, in fact.
An extensive simulation performance analysis of a simplified version of the APSS is carried out both in steady state and in dynamic response, running in normal operation and in protection modes.
Even though a comparison between experimental and simulated results is shown, it only meets the steady state; neither start-up process or battery charging schedule, two of the three key problems are not experimentally proven; I addition, the dynamic response or the protection mode in the test bench have not been contrasted.
Such an impact factor index journal should require more than almost a simulation study. The proposal lacks contrasting results, so the authors should be encouraged to broaden the experimental performance analysis at least and to resubmit their interesting proposal.
Regarding the Instructions for Author’s compliance:
Line #15: consider not to use a term of 8 nouns
Line #35: CHR abbreviation is not defined before
Line #92: please, revise if ‘start’ refers to a 3rd person singular in present tense
Line #94: please, revise if ‘start’ refers to a 3rd person singular in present tense once more
Line #134: please, revise if ‘control’ refers to a 3rd person singular in present tense
Line #136: please, revise if ‘reaching’ refers to a 3rd person singular in present tense
Line #138: the binomial expression of a complex value might start with the real part; so, please, revise if ‘jX1+r1’ should be better expressed as ‘r1+jX1’
Line #142: it might be a missed article before ‘current circulation’
Line #146: same as #138
Line #154: in eq.2, I*1 is not defined before use
Line #155: in eq.3, U1 is not defined before use
Line #156: please, revise if ‘is’ refers to a 3rd person plural in present tense
Line #158: it looks like a missing space after a full stop
Line #190: please, declare the axes units in fig.8
Line #188: what does ‘American scientist Mas’ mean?
Lines #189, 192 & fig.8 caption: especially, what’s the meaning of ‘Mas’?
Line #286: fig.17 is not referred to in the main text
Line #303: as in #158
Section #4.3: the comparison between experimental and simulated results should refer figures 19 and 16, because of an undeclared fig.17, only is performed to steady state; neither the start-up process or the battery charging schedule, two of the three key problems are not experimentally proven; in addition, the dynamic response in the test bench has not been contrasted.
Author Response
Dear reviewer:
I appreciate your attention on and response to this paper. Please review the following responses to referee comments in the file 2.
Thank you very much for all your help and looking forward to hearing from you soon.

Round 2
Reviewer 1 Report
The paper can be accepted in this form.